# Circulating Short-Chain Fatty Acids and Non-Alcoholic Fatty Liver Disease Severity in Patients with Type 2 Diabetes Mellitus

**DOI:** 10.3390/nu15071712

**Published:** 2023-03-31

**Authors:** Hui-Ju Tsai, Wei-Chun Hung, Wei-Wen Hung, Yen-Jung Lee, Yo-Chia Chen, Chun-Ying Lee, Yi-Chun Tsai, Chia-Yen Dai

**Affiliations:** 1Department of Family Medicine, Kaohsiung Municipal Ta-Tung Hospital, Kaohsiung Medical University, Kaohsiung 801, Taiwan; 2Department of Family Medicine, Kaohsiung Medical University Hospital, Kaohsiung Medical University, Kaohsiung 807, Taiwan; 3Department of Family Medicine, School of Medicine, College of Medicine, Kaohsiung Medical University, Kaohsiung 807, Taiwan; 4Research Center for Precision Environmental Medicine, Kaohsiung Medical University, Kaohsiung 807, Taiwan; 5Department of Microbiology and Immunology, College of Medicine, Kaohsiung Medical University, Kaohsiung 807, Taiwan; 6Division of Endocrinology and Metabolism, Kaohsiung Medical University Hospital, Kaohsiung 807, Taiwan; 7Center of Research Resources and Development, Kaohsiung Medical University, Kaohsiung 807, Taiwan; 8Department of Biological Science and Technology, National Pingtung University of Science and Technology, Pingtung 912, Taiwan; 9Division of General Medicine, Kaohsiung Medical University Hospital, Kaohsiung 807, Taiwan; 10Division of Nephrology, Kaohsiung Medical University Hospital, Kaohsiung 807, Taiwan; 11Center for Liquid Biopsy and Cohort Research, Kaohsiung Medical University, Kaohsiung 807, Taiwan; 12Hepatobiliary Division, Department of Internal Medicine, Kaohsiung Medical University Hospital, Kaohsiung Medical University, Kaohsiung 807, Taiwan

**Keywords:** non-alcoholic fatty liver disease, type 2 diabetes mellitus, short-chain fatty acids

## Abstract

(1) Background: Non-alcoholic fatty liver disease (NAFLD) is a major global health concern. The increasing prevalence of NAFLD has been related to type 2 diabetes mellitus (T2D). However, the relationship between short-chain fatty acids (SCFAs) and NAFLD severity is ambiguous in T2D subjects. This study aimed to explore the association of SCFAs with the severity of NAFLD in T2D patients. (2) Methods: We employed echography to examine the severity of hepatic steatosis. The serum levels of nine SCFAs, namely, formate, acetate, propionate, butyrate, isobutyrate, methylbutyrate, valerate, isovalerate, and methylvalerate, were measured using gas chromatography mass spectrometry. (3) Results: A total of 259 T2D patients was enrolled in this cross-sectional study. Of these participants, 117 with moderate to severe NAFLD had lower levels of formate, isobutyrate, and methylbutyrate than the 142 without NAFLD or with mild NAFLD. Lower circulating levels of isobutyrate and methylbutyrate were associated with an increased severity of NAFLD. A relationship between NAFLD severity and circulating isobutyrate and methylbutyrate levels was found independently of a glycated hemoglobin (HbA1C) level of 7.0%. (4) Conclusion: Circulating levels of isobutyrate and methylbutyrate were significantly and negatively correlated with NAFLD severity in the enrolled T2D patients. SCFAs may be related to NAFLD severity in T2D patients.

## 1. Introduction

Non-alcoholic fatty liver disease (NAFLD) is a crucial global public health concern, with growing prevalence over recent decades [1,2]. In Taiwan, after improvement in antiviral treatment for viral hepatitis, NAFLD has become one of the most common liver diseases [3,4]. NAFLD is defined as the pathologic accumulation of adipose tissue in the liver exceeding 5% of the total weight of the liver in the absence of significant alcohol consumption [5], and it can progress to liver cirrhosis and hepatocellular carcinoma [6]. NAFLD is an intricate disorder mediated by metabolic, environmental, polygenic, and microbiologic mechanisms [2]. The increasing prevalence of NAFLD has been linked to metabolic diseases such as type 2 diabetes mellitus (T2D), hypertension, chronic kidney disease, obesity, and hyperlipidemia [7]. NAFLD and T2D commonly coexist because they share similar pathophysiologic mechanisms, such as insulin resistance, genetic predisposition, and environmental factors [8,9].

The gut microbiota has frequently been linked to metabolic diseases such as NAFLD, diabetes, and obesity, and several studies have reported that dysregulation of the intestinal microbiota (gut dysbiosis) can cause NAFLD [10,11]. The gut microbiota and its metabolites have been found to affect multiple physiological mechanisms related to human health. Short-chain fatty acids (SCFAs) are volatile fatty acids produced by intestinal bacteria to metabolize dietary fiber. Acetic acid, propionic acid, and butyric acid are the most abundant, representing 90–95% of the SCFAs present in the colon. The main sources of SCFAs are carbohydrates; however, amino acids such as valine, leucine, and isoleucine produced by protein breakdown can be transformed into isobutyrate, isovalerate, and 2-methyl butyrate, known as branched-chain fatty acids (BCFAs), which account for 5% of the total SCFA production [12,13]. Acetic acid and propionate are mostly produced by Bacteroidetes, whereas butyrate is principally produced by Firmicutes [14]. SCFAs involve glucose metabolism, insulin sensitivity, and lipogenesis through diverse pathways, thereby affecting the development of diabetes, obesity, and NAFLD [15,16]. Furthermore, increasing human evidence has shown the beneficial effect of SCFAs on body weight control, inflammatory status, and insulin sensitivity, as well as on glucose and lipid homeostasis [17,18,19]. However, the impact of SCFAs on the severity of NAFLD has not been well explored, especially in T2D patients. Therefore, the aim of this study was to explore the association between the severity of NAFLD and the circulating SCFA levels in T2D patients.

## 2. Materials and Methods

### 2.1. Study Subjects

A total of 259 patients with T2D were recruited between October 2016 and December 2017 in this observational research from a tertiary hospital in the south of Taiwan. T2D was defined by prescriptions for antidiabetic drugs, a history of diabetes, or blood glucose values, according to the criteria of the American Diabetes Association. The exclusion criteria were patients who consumed an average of >30 g/day of alcohol and the presence of autoimmune hepatitis or hepatitis B and C. All patients were asked to attend T2D education programs. The Institutional Review Board of Kaohsiung Medical University Hospital (KMUHIRB-G(II)-20160021) gave ethical approval for the study, which was conducted according to the Declaration of Helsinki, and all enrollees provided written informed consent.

### 2.2. Sample and Clinical Information Collection

Demographic data, including a history of tobacco smoking and alcohol consumption, and clinical data were collected through interviews and medical records at enrollment. The medical records were also used to collect information on medication usage, including statins (one of antihyperlipidemic agents) and antidiabetic drugs such as sulfonylurea, Dipeptidyl peptidase 4 (DPP4) inhibitor, metformin, thiazolidinediones, and insulin. The patients recorded their dietary habits using a basic questionnaire. Body mass index (BMI) was recorded as kg/m^2^. Blood and urine samples were collected after a 12 h fast for biochemical studies and to measure albuminuria, respectively.

### 2.3. Measurement of NAFLD

A single operator blinded to the patients’ status performed echography to assess the severity of the NAFLD. Normal liver parenchyma was defined as that with equivalent or slightly greater echogenicity than the adjacent spleen and kidney. Lipid droplets in steatosis scatter ultrasound beams, resulting in the transducer receiving more echo signals and consequently causing the appearance of a hyperechoic or “bright” liver. Fat also reduces the strength of the beam, thereby decreasing its penetration into tissues. The brightness of the liver and/or dimming of the vessels and diaphragm can therefore be used to assess the severity of steatosis. To avoid inaccuracies caused by factors associated with acquisition, the brightness of the liver was estimated through comparison with the spleen or kidney as a standard reference [20]. The degree of fatty liver was then classified as none, mild, moderate, or severe.

### 2.4. Measurement of SCFAs

The serum levels of nine SCFAs, namely, formate, acetate, propionate, butyrate, isobutyrate, methylbutyrate, valerate, isovalerate, and methylvalerate, were measured using liquid chromatography mass spectrometry (LC–MS/MS). Human serum (50 μL) was mixed with 20 μL of 3-Nitrophenylhydrazine hydrochloride (200 mM) in 100% aqueous methanol and 20 μL of 120 mM *N*-(3-Dimethylaminopropyl)-*N*′-ethylcarbodiimide hydrochloride-6% pyridine solution in 100% aqueous methanol. The mixture was reacted at 40 °C for 30 min. The solution was then diluted to 210 μL with 10% aqueous methanol. A 75 μL aliquot was mixed with 25 μL of the internal standard (IS) mix solution, and a 10 μL aliquot was injected for LC–MS/MS.

An ultraperformance liquid chromatography system (ACQUITY, Waters Corporation, Milford, MA, USA) with tandem MS (Finnigan TSQ Quantum Ultra triple-quadrupole MS, Thermo Electron, San Jose, CA, USA) and Xcalibur software 2.2 (ThermoFinnigan, Bellefonte, PA, USA) were used for the detection and quantification analysis. The LC–MS/MS system was equipped with an electrospray ion source and was run in positive mode. A volume of 10 μL was injected into an ultraperformance liquid chromatography column (ACQUITY UPLC BEH C18, 130 Å, 1.7 µm, 2.1 mm × 100 mm, Waters Corporation) equipped with a filter in front of the column. The flow rate was 300 μL/min and the column temperature was 40 °C.

### 2.5. Statistical Analysis

The T2D participants were categorized on the basis of the severity of NAFLD to a no NAFLD and mild NAFLD group and a moderate and severe NAFLD group, and their baseline characteristics were compared. If continuous data are normally distributed, they are presented as a mean ± SD. If continuous data are non-normally distributed, they are presented as a median (interquartile range). Categorical data are presented as a percentage. Non-normally distributed continuous variables were log-transformed for normal distribution. Between-group differences in the continuous variables were compared using the independent *t*-test or Mann–Whitney U analysis, as appropriate, and the chi-square test was used for categorical variables. To reduce the effect of possible confounding factors on the relationship between circulating SCFA levels and the severity of NAFLD, the covariates, including clinical data, laboratory data, and medication, were initially analyzed with univariate analysis. Then multivariate logistic regression models, including significant variables (*p* < 0.05) in univariate analysis, and other traditional covariates, such as habit of smoking, alcohol drinking, and medical disease, were adjusted to clarify the association between circulating SCFA levels and the severity of NAFLD. We further categorized the patients according to sex, obesity (defined as a BMI of 27 kg/m^2^), and a glycated hemoglobin (HbA1C) level of 7% to further evaluate the synergistic interactions between the SCFAs and sex, obesity, and sugar control in the severity of NAFLD. Statistical analyses were conducted using SPSS version 22.0 for Windows (IBM Inc., Armonk, NY, USA) and GraphPad Prism version 9.0 (GraphPad Software Inc., San Diego, CA, USA). Statistical significance was set at a two-sided *p*-value smaller than 0.05.

## 3. Results

### 3.1. Demographic and Clinical Data of the Study Population

Comparison of the demographic and clinical characteristics between the two groups divided by the severity of NAFLD is described in Table 1. Of the 259 enrolled T2D patients, 142 had no NAFLD or mild NAFLD and 117 had moderate to severe NAFLD. The mean age of the patients was 61.4 ± 10.6 years, the age ranged from 25.4 to 88.1 years, and the mean duration of T2D was 10.2 ± 8.3 years. Of all subjects, 151 male and 108 female patients were included; 21.7% habitually drank alcohol and 27.9% smoked. The prevalence rates of hypertension, gout, and hyperlipidemia were 60.6%, 11.6%, and 80.7%, respectively. The usage rate of sulfonylurea, DPP4 inhibitor, metformin, thiazolidinediones, insulin, and a statin was 40.5%, 65.6%, 74.9%, 27.4%, 12.7%, and 45.9%, respectively.

The age of the subjects with moderate to severe NAFLD ranged from 25.8 to 82.5 years, and the age of the subjects without NAFLD or with mild NAFLD ranged from 35.8 to 88.2 years. Therefore, the subjects with moderate to severe NAFLD were younger than the subjects without NAFLD or with mild NAFLD. The subjects with moderate to severe NAFLD also had a higher BMI than the subjects without NAFLD or with mild NAFLD. There were no significant differences in the proportion of smoking, drinking alcohol, or usage of sulfonylurea, DPP4 inhibitor, metformin, thiazolidinediones, insulin, or statin between the two groups. The moderate to severe NAFLD group had higher glutamic oxaloacetic transaminase (GOT), glutamic pyruvic transaminase (GPT), triglyceride (TG), and HbA1C levels than those without NAFLD or with mild NAFLD.

### 3.2. The Composition of Serum SCFAs in T2D Subjects with Different NAFLD Severity

We investigated the serum levels of formate, acetate, propionate, butyrate, isobutyrate, methylbutyrate, valerate, isovalerate, and methylvalerate using LC–MS in the study patients. The T2D subjects with moderate to severe NAFLD had lower levels of circulating formate, isobutyrate, and methylbutyrate than the subjects without NAFLD or with mild NAFLD. Higher isovalerate levels were borderline significantly found in T2D subjects with moderate to severe NAFLD. There were no significant differences in circulating acetate, propionate, butyrate, valerate, or methylvalerate between the two groups (Table 1 and Figure 1).

### 3.3. Circulating SCFA Levels and the NAFLD Severity in T2D Subjects

To explore the association between the circulating SCFA levels and the severity of NAFLD in T2D patients, the logistic regression model was used and clinical data, laboratory data, and medications were also analyzed (Table 2). In univariate analysis, BMI ≥ 27 kg/m^2^; metformin usage; and serum GOT, GPT, hemoglobin (Hb), log-formed TG, and isovalerate levels were significantly and positively correlated with an elevated risk of moderate to severe NAFLD in T2D patients (Table 2). Age and plasma isobutyrate and methylbutyrate levels were negatively associated with an elevated risk of moderate to severe NAFLD. After adjusting for age, BMI ≥ 27 kg/m^2^, T2D duration, Hb, GOT, GPT, log-formed TG, metformin usage, habit of smoking, alcohol drinking, history of hypertension, gout, and hyperlipidemia, the T2D patients with low levels of circulating isobutyrate (odds ratio (OR): 0.17, 95% confidence index (CI): 0.03–0.86) and methylbutyrate (OR: 0.25, 95% CI: 0.08–0.76) levels had an increased risk of moderate to severe NAFLD. However, there was no significant relationship between circulating isovalerate and the severity of the NAFLD in the adjusted analysis.

In order to investigate the impacts of sex, obesity, and sugar control on the relationship between circulating isobutyrate and methylbutyrate levels and the severity of NAFLD, we categorized the participants on the basis of sex, BMI of 27 kg/m^2^, and HbA1C of 7.0%, and the results of the relationships between circulating isobutyrate and methylbutyrate levels and the severity of NAFLD independently of glycemic control remained consistent (Figure 2A,B). Negative correlations between circulating isobutyrate and methylbutyrate levels and the severity of NAFLD were found in men but not in women. In addition, a negative correlation between circulating methylbutyrate levels and the severity of NAFLD was found in subjects with a BMI ≥ 27 kg/m^2^.

## 4. Discussion

This research investigated the relationship of SCFAs with the severity of fatty liver in T2D subjects. Our findings showed low levels of circulating isobutyrate and methylbutyrate in the subjects with moderate to severe NAFLD in comparison with other subjects without NAFLD or with mild NAFLD. Therefore, the patients with low circulating isobutyrate and methylbutyrate levels had an increased risk of more severe NAFLD. After reviewing the literature, this is the first study to evaluate the impact of circulating SCFA levels on the severity of NAFLD in subjects with T2D. SCFAs may not only have an impact on the pathophysiology of NAFLD, but also be independently linked to the severity of NAFLD in T2D subjects.

Previous studies have reported a relationship between SCFAs and the pathophysiology of NAFLD [21,22,23]. Gut dysfunction, including dysbiosis, alterations in SCFAs, and increased gut permeability, are thought to promote the progression of NAFLD from the relatively benign hepatic steatosis toward non-alcoholic steatohepatitis [24,25]. We found that the patients with low circulating isobutyrate and methylbutyrate levels had an elevated risk of more severe NAFLD. Both isobutyrate and methylbutyrate are BCFAs, which have one or more methyl branches on the carbon chain. Few studies have investigated the association between BCFAs and metabolic disorders in humans. Heimann et al. observed that BCFAs had effects on adipocyte lipids and glucose metabolism that could contribute to improved insulin sensitivity in individuals with a disturbed metabolism [26]. In addition, Pakiet et al. found an inverse correlation between serum BCFAs and the homeostasis model assessment-insulin resistance index (HOMA-IR index) in 82 participants, suggesting that BCFAs may promote insulin sensitivity [27]. Moreover, another study found that obese individuals had lower circulating isobutyrate levels than healthy individuals [28]. Su et al. also found that the adipose tissue of lean individuals had higher proportions of total BCFAs and individual BCFAs than that of obese subjects [29]. However, a few human observational studies have reported that fecal BCFA concentrations were higher in individuals with obesity [30] and hypercholesterolemia [31], and that this was associated with NAFLD progression [32]. We found a relationship between the circulating level of methylbutyrate and the severity of NAFLD only in the obese participants. In view of these inconsistent findings, it is important to consider that the impact of serum SCFA and BCFA concentrations on NAFLD may depend on various factors including age, diet, microbial community, lifestyle, and co-existing diseases [33,34].

SCFAs including butyrate, acetate, and propionate have been reported to improve hepatic steatosis either by activating G protein-coupled receptor 41 (GPR41) and GPR43 receptors in adipose tissue, the intestine, and the liver, or by directly acting locally without binding to their receptors [23]. In addition, SCFAs are carried to the liver by the portal circulation, thereby serving as precursors for gluconeogenesis and lipogenesis [35,36]. However, we did not find significant associations between butyrate, acetate, and propionate and the severity of NAFLD. BCFAs are a major component of the membranes of many bacteria, including Lactobacilli and Bifidobacteria [37]. A few studies have suggested that Bifidobacteria may have a protective effect against the development of NAFLD and obesity [38,39]. We also found that the patients with lower levels of BCFAs, including isobutyrate and methylbutyrate, had a higher risk of more severe NAFLD.

In this study, we also explored the interaction effect of glycemic control on the relationship between circulating isobutyrate and methylbutyrate levels and the severity of NAFLD in T2D subjects. Our results showed negative relationships between the severity of NAFLD and circulating isobutyrate and methylbutyrate levels independent of glycemic control. Previous studies have reported that SCFAs act as mediators between the gut microbiota and the pancreas directly through receptors on pancreatic cells or via the gut-brain-pancreatic axis [15]. SCFAs have the capacity not only to improve glucose homeostasis and insulin sensitivity, but also to regulate pancreatic insulin and glucagon secretion through glucagon-like peptide-1 augmentation in pancreatic dysfunction [15,40]. The pathophysiologic overlap between NAFLD and T2D may complicate the establishment of a specific bacterial signature related to the disease spectrum.

In addition, we found that the circulating levels of isobutyrate and methylbutyrate were negatively correlated with the severity of NAFLD in the male patients. Whether there are sex differences in the association between SCFAs and NAFLD severity is unknown. A previous study suggested that men have lower SCFA levels and dietary fiber intake compared with women [41]. Sex differences in the prevalence and severity of NAFLD have been shown. Differences in biological factors and lifestyle between men and women may suggest that there are also sex differences in the relationship between SCFAs and NAFLD.

Some studies have measured SCFA in feces because the concentration range of SCFAs in the colon is at the millimolar level. However, this may not accurately reflect direct interactions with related organs, because approximately 95% of colonic SCFAs are absorbed, and only the remaining 5% are excreted in the feces [42]. Because of the rapid turnover rate and dynamic variations in the blood, the concentration of SCFAs in the blood is at the micromolar level, which is much lower than the concentration in the feces [43]. Despite the low concentration of SCFAs in serum, serum SCFAs are considered to be more strongly linked with metabolic disease than fecal SCFAs because they directly interact with the target tissues and organs through their receptors [34]. One study found that serum SCFAs, rather than fecal SCFAs, were associated with metabolic markers such as glucagon-like peptide 1, fatty acid metabolism, and insulin sensitivity [44]. Therefore, the relationship between SCFAs and metabolic disease may be inconsistent depending on their source.

Previous studies have suggested that antidiabetic drugs may influence gut microbiota composition and the biosynthesis of SCFAs [45]. In this study, the antidiabetic drugs including sulfonylurea, DPP4 inhibitor, metformin, thiazolidinediones, and insulin were considered. Univariate analysis showed only metformin usage was positively related to the severity of NAFLD in T2D patients. However, after adjusting for the usage of metformin, the negative correlation between the circulating levels of isobutyrate and methylbutyrate and NAFLD severity in the T2D patients was still found. Furthermore, we found that other antidiabetic drugs, including sulfonylurea, DPP4 inhibitor, thiazolidinediones, and insulin, were not significantly related to NAFLD severity. Metformin regulates glucose uptake, gluconeogenesis, glycolysis, and glycogen synthesis in the liver, and modifies bile acid recirculation in the gut [45,46]. In addition, metformin may accelerate SCFA-producing bacteria, such as Blautia, Bacteroides, Bifidobacterium, and Prevotella [45]. Metformin may also increase the abundance of mucin-degrading bacteria, such as Lactobacillus and Akkermansia [45]. There is a lack of information about the effect of antidiabetic drugs on the NAFLD severity in T2D patients. Moreover, the relationship between antidiabetic drugs and SCFAs is also unclear. Further studies are needed to investigate the interaction between antidiabetic drugs, SCFAs, and NAFLD severity.

This study has several limitations. First, SCFAs are mostly derived from the microbial fermentation of dietary fiber. However, we did not have detailed information on dietary fiber intake. Nevertheless, we included the crude dietary habits of the patients to minimize the impact of diet on the relationship between SCFAs and NAFLD. Second, there were no nondiabetic individuals in this study to compare with the T2D patients. However, our study aim was to examine the relationship between circulating SCFAs and NAFLD severity in T2D patients, not in the general population. Thus, these findings were not influenced regardless of whether nondiabetic individuals were enrolled in this study. Third, this cross-sectional study recruited a comparatively small sample size of participants, and the causality cannot be identified. In addition, we did not estimate SCFAs in the stool to compare the differences with serum SCFAs. Further research is needed to explore the interactions among SCFAs in the serum or feces, microbiota, and the severity of NAFLD in T2D subjects. Finally, the biological mechanism is clearly unknown, and future in vitro and in vivo researches are necessary to investigate the pathophysiologic mechanism of SCFAs in NAFLD progression and other novel biomarkers in T2D patients.

## 5. Conclusions

In conclusion, circulating levels of isobutyrate and methylbutyrate were significantly and negatively correlated with NAFLD severity in the enrolled T2D subjects. Apart from the usual metabolic determinants, SCFAs may be associated with NAFLD severity in T2D patients.

## Figures and Tables

**Figure 1 nutrients-15-01712-f001:**
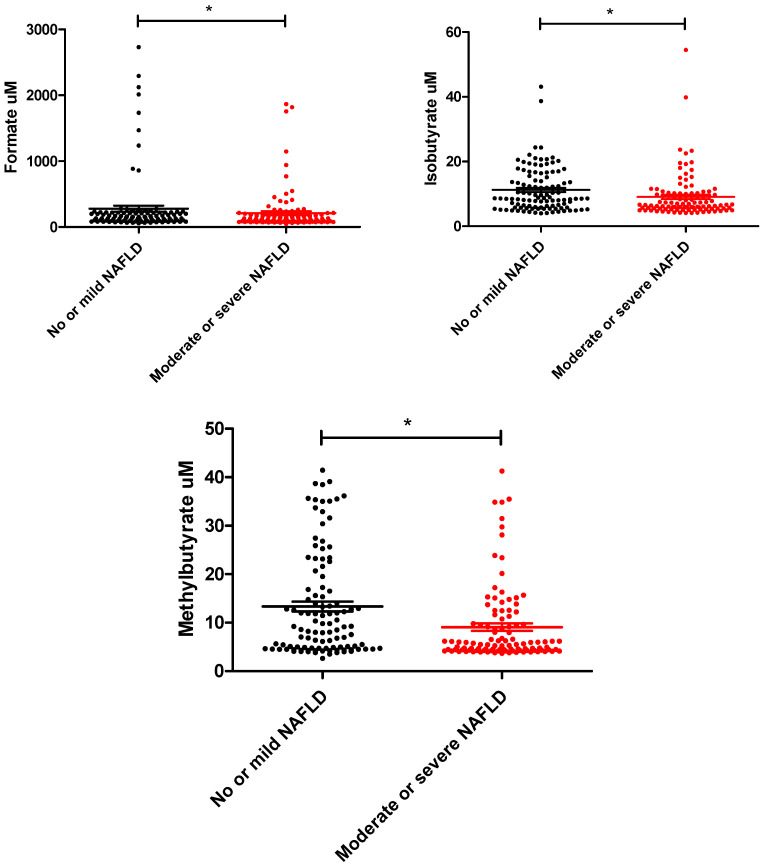
The distribution of circulating levels of formate, isobutyrate, and methylbutyrate between T2D patients with no–mild and mild–moderate NAFLD. (*****
*p* value < 0.05)

**Figure 2 nutrients-15-01712-f002:**
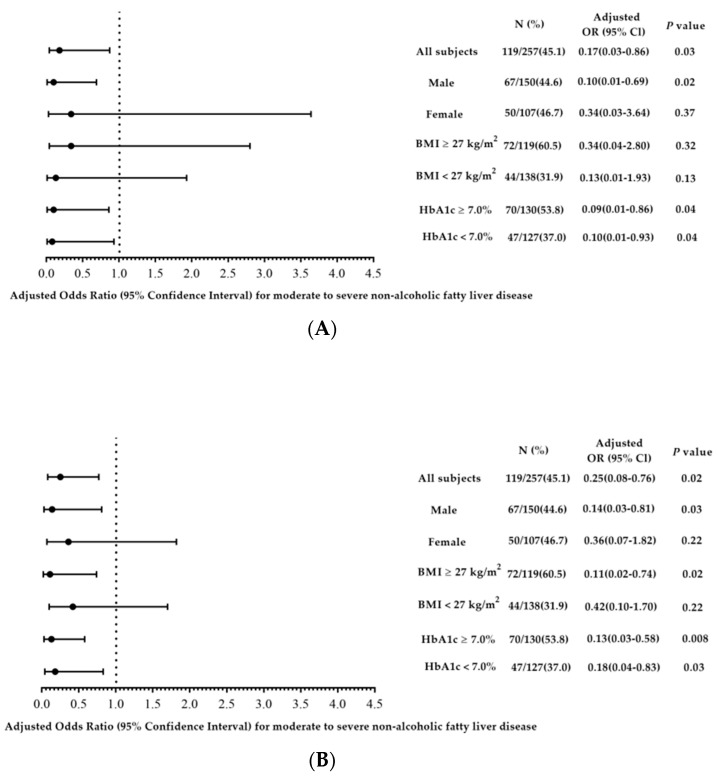
Adjusted odds ratios (ORs) of log-formed (**A**) isobutyrate and (**B**) methylbutyrate with the severity of NAFLD in T2D subjects divided by sex, BMI cut at 27 kg/m^2^, and HbA1C of 7%. ORs were adjusted for age, BMI ≥ 27 kg/m^2^, T2D duration, Hb, log-formed TG, GOT, GPT, metformin usage, habit of smoking, alcohol drinking, history of hypertension, gout, and hyperlipidemia.

**Table 1 nutrients-15-01712-t001:** The demographic and clinical data of the study participants.

	All Participants	No or Mild NAFLD	Moderate or Severe NAFLD	*p* Value
Number	259	142	117	
Age, years	61.4 ± 10.6	63.6 ± 10.9	58.8 ± 9.5	0.03
	(25.4–88.1)	(35.8–88.2)	(25.8–82.5)	
Sex, %				0.76
Male	58.3	59.2	57.3	
Female	41.7	40.8	42.7	
Habit of smoking, %	27.9	28.4	27.4	0.85
Alcohol drinking, %	21.7	18.4	25.6	0.16
Hypertension, %	60.6	63.4	57.3	0.31
Gout, %	11.6	13.4	9.4	0.32
Hyperlipidemia, %	80.7	78.9	82.9	0.41
DM duration, years	10.2 ± 8.3	10.9 ± 8.7	7.5 ± 7.2	0.002
BMI, kg/m^2^	26.6 ± 4.4	25.7 ± 4.0	28.9 ± 4.5	<0.001
BMI ≥ 27 kg/m^2^	46.3	33.3	62.1	<0.001
Medication				
Sulfonylurea (%)	40.5	39.4	41.9	0.69
DPP4 inhibitor (%)	65.6	63.4	68.4	0.40
Metformin (%)	74.9	69.0	82.1	0.02
Thiazolidinediones (%)	27.4	29.6	24.8	0.39
Insulin (%)	12.7	14.1	11.1	0.47
Statin (%)	45.9	45.8	46.2	0.95
Short-chain fatty acid	Median (25th, 75th percentile)			
Formate	125.0 (89.7,214.4)	141.8 (97.5, 218.5)	105.7 (82.6, 171.8)	0.01
Acetate	97.4 (74.7, 133.6)	89.2 (71.7, 125.7)	97.1 (74.0, 137.6)	0.33
Propionate	15.4 (11.7, 21.4)	15.0 (11.5, 21.7)	14.5 (10.8, 20.7)	0.49
Butyrate	8.1 (6.1, 9.8)	7.9 (5.4, 9.7)	8.1 (7.1, 9.3)	0.43
Isobutyrate	7.6 (5.4, 12.5)	8.6 (5.8, 15.4)	6.6 (5.4, 10.2)	0.003
Methylbutyrate	6.2 (4.5, 13.5)	9.2 (4.7, 18.9)	5.6 (4.3, 10.9)	0.001
Valerate	2.7 (1.7, 5.0)	2.8 (1.6, 5.7)	2.5 (1.6, 4.6)	0.33
Isovalerate	17.8 (4.0, 24.9)	8.1 (3.3, 22.9)	18.9 (4.9, 22.9)	0.04
Methylvalerate	1.4 (0.7, 3.3)	1.6 (0.7, 3.7)	1.5 (0.8, 2.6)	0.45
Laboratory parameters	Mean ± SD or median (25th, 75th percentile)			
Hb (g/dL)	13.7 ± 1.7	13.2 ± 1.8	14.3 ± 1.7	<0.001
UA (mg/dL)	5.9 ± 1.6	6.0 ± 1.7	6.0 ± 1.6	0.93
GOT (U/L)	30.0 ± 14.5	29.7 ± 15.3	38.8 ± 20.9	<0.001
GPT (U/L)	32.8 ± 23.8	31.9 ± 26.0	48.7 ± 34.5	<0.001
Creatinine (mg/dL)	1.0 ± 0.5	1.2 ± 0.7	0.9 ± 0.3	<0.001
Cholesterol (mg/dL)	170.4 ± 40.6	171.8 ± 47.4	174.7 ± 40.5	0.60
Triglyceride (mg/dL)	120.0 (85.7, 179.3)	93 (65, 140)	133 (100, 186)	<0.001
HDL (mg/dL)	45.2 ± 17.4	44.2 ± 12.5	44.5 ± 26.6	0.92
LDL (mg/dL)	96.3 ± 33.3	98.3 ± 38.8	98.5 ± 32.9	0.96
HbA1C (%)	7.0 (6.5, 8.0)	6.8 (6.4, 7.5)	7.0 (6.5, 8.0)	0.01

Abbreviations: NAFLD, nonalcoholic fatty liver disease; BMI, body mass index; DPP4, Dipeptidyl peptidase 4; Hb, hemoglobin; UA, uric acid; GOT, glutamate oxaloacetate transaminase; GPT, glutamate pyruvate transaminase; HDL, high-density lipoprotein; LDL, low-density lipoprotein.

**Table 2 nutrients-15-01712-t002:** The relationship between these determinants and the NAFLD severity in logistic regression analysis.

Moderate to Severe NAFLD	Crude OR (95%Cl)	*p*-Value	Adjusted OR (95%Cl)	*p*-Value	Adjusted OR (95%Cl)	*p*-Value
Clinical characteristics						
Age, years	0.97 (0.95–0.99)	0.01	0.97 (0.93–1.01)	0.16	0.97 (0.93–1.01)	0.17
Sex (female vs. male)	1.08 (0.66–1.77)	0.76	-	-	-	-
BMI ≥ 27 kg/m^2^	1.20 (1.12–1.28)	<0.001	2.35 (1.09–5.07)	0.03	2.30 (1.06–4.97)	0.03
Habit of smoking (yes vs. no)	0.95 (0.55–1.64)	0.85	1.15 (0.44–3.02)	0.78	1.12 (0.43–2.94)	0.81
Alcohol drinking (yes vs. no)	1.53 (0.84–2.76)	0.16	1.07 (0.38–3.07)	0.89	1.12 (0.39–3.21)	0.83
T2D duration, years	0.94 (0.91–0.98)	0.004	0.99 (0.95–1.04)	0.67	0.99 (0.95–1.04)	0.68
Hypertension (yes vs. no)	0.77 (0.45–1.28)	0.32	0.80 (0.38–1.68)	0.55	0.84 (0.39–1.77)	0.64
Gout (yes vs. no)	0.67 (0.31–1.48)	0.32	1.10 (0.32–3.84)	0.88	1.05 (0.30–3.73)	0.94
Hyperlipidemia (yes vs. no)	1.30 (0.69–2.43)	0.41	1.46 (0.54–3.95)	0.46	1.51 (0.56–4.10)	0.42
Laboratory data						
UA (mg/dL)	0.99 (0.86–1.15)	0.93	-	-	-	-
Hb (g/dL)	1.44 (1.24–1.68)	<0.001	1.16 (0.90–1.51)	0.26	1.16 (0.89–1.50)	0.26
GOT (U/L)	1.03 (1.01–1.05)	<0.001	1.01 (0.97–1.06)	0.55	1.01 (0.97–1.06)	0.58
GPT (U/L)	1.02 (1.01–1.03)	<0.001	1.00 (0.97–1.03)	0.94	1.00 (0.97–1.03)	0.96
Cholesterol (mg/dL)	1.00 (0.99–1.01)	0.60	-	-	-	-
Log (Triglyceride)	1.01 (1.00–1.01)	<0.001	4.05 (0.72–22.93)	0.11	4.21 (0.74–23.92)	0.10
HDL (mg/dL)	1.00 (0.99–1.01)	0.92	-	-	-	-
LDL (mg/dL)	1.00 (0.99–1.01)	0.96	-	-	-	-
HbA1C (%)	1.12 (0.96–1.31)	0.15	-	-	-	-
Medication						
Sulfonylurea (yes vs. no)	1.11 (0.67–1.82)	0.69	-	-	-	-
DPP4 inhibitor (yes vs. no)	1.25 (0.74–2.10)	0.40	-	-	-	-
Metformin (yes vs. no)	2.05 (1.14–3.71)	0.02	2.27 (0.79–6.48)	0.13	2.27 (0.80–6.47)	0.13
Thiazolidinediones (yes vs. no)	0.79 (0.45–1.36)	0.39	-	-	-	-
Insulin (yes vs. no)	0.76 (0.36–1.61)	0.48	-	-	-	-
Statin (yes vs. no)	1.02 (0.62–1.66)	0.95	-	-	-	-
SCFA						
Log (Formate)	0.47 (0.20–1.07)	0.07	-	-		
Log (Acetate)	2.28 (0.54–9.60)	0.26	-	-		
Log (Propionate)	0.79 (0.22–2.79)	0.71	-	-		
Log (Butyrate)	1.66 (0.41–6.66)	0.48	-	-		
Log (Isobutyrate)	0.16 (0.05–0.56)	0.004	0.17 (0.03–0.86)	0.03		
Log (Methylbutyrate)	0.21 (0.08–0.53)	0.001	-	-	0.25 (0.08–0.76)	0.02
Log (Valerate)	0.66 (0.31–1.40)	0.28	-	-		
Log (Isovalerate)	2.09 (1.14–3.82)	0.02	-	-		
Log (Methylvalerate)	0.77 (0.40–1.50)	0.44	-	-		

Abbreviations: OR, odds ratio; other variables as abbreviations in Table 1.

## Data Availability

The data presented in this study are available on request from the corresponding author. The data are not publicly available due to privacy.

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
