# Peer review of "Circulating Short-Chain Fatty Acids and Non-Alcoholic Fatty Liver Disease Severity in Patients with Type 2 Diabetes Mellitus"

_nutrients, 2023, doi:10.3390/nu15071712_

Round 1

Reviewer 1 Report

The authors have highlighted the relationship between short-chain fatty acids and NAFLD severity in type 2 diabetic patients. No doubt, it is a significant point to explore however, the study has some limitations in addition to those highlighted by the authors at the end of the manuscript.

The study has no non-diabetic controls to compare the level of SCFs in the people who smoke, drink alcohol, have hypertension or hyperlipidemia but do not have type 2 diabetes. Non-diabetic controls could provide good comparisons if the diet of the patients was not controlled.  

The authors mentioned low levels of formate, isobutyrate, and methylbutyrate associated with NAFLD severity. Do high levels of acetate and isovalerate also relate to NAFLD severity as shown in table 1?

Authors must provide more information about the patients, for example, how many males and females, and how many different types of diseases the patients had in addition to T2D. Were they have a comparable level of T2D and were using the same antidiabetic drug for T2D? This is well known that gut microbial species change with antibiotics treatment and could have a different impact on the overall health of humans depending on the type of drug used (doi: 10.3390/genes8100250).  

All T2D patients do not have NAFLD. Could it be concluded that the risk of NAFLD is not related to T2D, the risk of NAFLD in T2D patients is related to a specific medication, lifestyle, food preferences, gender, or some other diseases in addition to T2D that alter gut microbiota and their metabolites (SCFs)?  If so, the conclusion of the study must be changed.

It would be worth adding some suggestions for future studies on the basis of your observation.

Please explain the points highlighted in red in the manuscript and remove some language errors. 

Author Response

The authors have highlighted the relationship between short-chain fatty acids and NAFLD severity in type 2 diabetic patients. No doubt, it is a significant point to explore however, the study has some limitations in addition to those highlighted by the authors at the end of the manuscript.

Question 1: The study has no non-diabetic controls to compare the level of SCFAs in the people who smoke, drink alcohol, have hypertension or hyperlipidemia but do not have type 2 diabetes. Non-diabetic controls could provide good comparisons if the diet of the patients was not controlled.  

Answer 1: Thanks for your excellent suggestion. This study focused on the relationship between circulating SCFA levels and NAFLD severity in T2D patients; thus, we don’t enroll non-diabetic controls. As reviewer’s concern, some life habit and co-existing diseases had the impacts on NAFLD severity. We have added this limitation in this manuscript. Please see the Discussion, Page 9, paragraph 5.

Question 2: The authors mentioned low levels of formate, isobutyrate, and methylbutyrate associated with NAFLD severity. Do high levels of acetate and isovalerate also relate to NAFLD severity as shown in table 1?

Answer 2:  Thanks for your suggestion. Higher isovalerate level was found in T2D patients with moderate to severe NAFLD. There were no significant differences in circulating acetate, propionate, butyrate, valerate, or methylvalerate between the two groups. We have revised it. Please see Page 4, paragraph 2

Question 3: Authors must provide more information about the patients, for example, how many males and females, and how many different types of diseases the patients had in addition to T2D. Were they have a comparable level of T2D and were using the same antidiabetic drug for T2D? This is well known that gut microbial species change with antibiotics treatment and could have a different impact on the overall health of humans depending on the type of drug used (doi: 10.3390/genes8100250).  

Answer 3: Thanks for your excellent suggestion. Sex, other diseases, and the anti-diabetic drugs were described in Table 1 and results. We also added the anti-diabetic drugs and statin as adjusted variables, and re-analyze the multiple logistic regression in Table 2 and Figure 2. Although metformin was significantly associated with NAFLD severity in univariate regression analysis, multiple logistic regression analysis still showed the significant relationship between isobutyrate and methylbutyrate levels and the severity of NAFLD after adjusting age, body mass index ≥ 27 kg/m2, T2D duration, metformin usage, hemoglobin, log-formed triglyceride, glutamate oxaloacetate transaminase, and glutamate pyruvate transaminase. We have revised it in the results and Table 1, Table 2, and Figure 2 (Page 4-7).We also have added description and cited this reference in the discussion. Please see page 9, paragraph 4, reference 45-46

Question 4: All T2D patients do not have NAFLD. Could it be concluded that the risk of NAFLD is not related to T2D, the risk of NAFLD in T2D patients is related to a specific medication, lifestyle, food preferences, gender, or some other diseases in addition to T2D that alter gut microbiota and their metabolites (SCFs)?  If so, the conclusion of the study must be changed.

Answer 4: Thanks for your suggestion. This study is aimed to evaluate the relationship between the severity of NAFLD and circulating SCFA levels in T2D patients. Specific medication, lifestyle, food preferences, gender, and co-existing diseases are related to the risk of NAFLD in T2D. Therefore, we collected these information including sex, habit of smoke, alcohol use, hypertension, hyperlipidemia, gout, and the medication including sulfonylurea, DPP4 inhibitor, metformin, thiazolidinediones, insulin and statin and analyzed and adjusted these confounding factors to evaluate the relationship between the severity of NAFLD and circulating SCFA levels in T2D patients. Please see results (Page 4 and page 6)

Question 5: It would be worth adding some suggestions for future studies on the basis of your observation.

 Answer 5: Thanks for your suggestion. We have added some suggestions for future studies in discussion. The biological mechanism is clearly unknown, and future in vitro and in vivo studies can be designed to identify the detailed mechanism between SCFAs and the severity of NAFLD in T2D patients. Please see page 9, paragraph 4 and page 10, paragraph 1

Question 6: Please explain the points highlighted in red in the manuscript and remove some language errors. 

Answer 6: Thanks for your suggestions. We have revised it and please see these changes with blue color.

Reviewer 2 Report

The investigators evaluated the relationship between the severity of NAFLD and various circulating short chain fatty acid levels in subjects with type-2 diabetes.

Although, some correlations were made between the SCFs and NAFLD in diabetic subjects, I feel the scientific and clinical relevance of the research is low. A mere association of a biomarker is not relevant for forming a diagnosis or study the prognosis of disease. The authors also could not give any future directions of the present study.

Author Response

Reviewer 2’s comments:

Question 1: The investigators evaluated the relationship between the severity of NAFLD and various circulating short chain fatty acid levels in subjects with type-2 diabetes.
Although, some correlations were made between the SCFs and NAFLD in diabetic subjects, I feel the scientific and clinical relevance of the research is low. A mere association of a biomarker is not relevant for forming a diagnosis or study the prognosis of disease. The authors also could not give any future directions of the present study.

Answer 1: Thanks for your comment. We have revised this manuscript and added further direction. The biological mechanism is clearly unknown, and future in vitro and in vivo studies can be designed to identify the detailed mechanism between SCFAs and the severity of NAFLD in T2D patients. Please see page 9, paragraph 4 and page 10, paragraph 1

Round 2

Reviewer 1 Report

The authors have revised the points mentioned in the comments however, the mixing of several parameters has made it difficult to believe in the conclusion. The participants had smoking and drinking habits and had a mixture of diseases that could be the reason for gut dysbiosis and alteration in SCFs regulation. The use of different drugs is another parameter which is not normalized in all participants. The authors have mentioned the lack of controls in the limitations of the study that indicates the potential false conclusion.  

Moreover, have a look at two minor points mentioned in the manuscript

1.      Write the full word of any abbreviation when use first time in the manuscript. Like SI mix solution (Does IS mean internal standard or International standard?). An expert can know however, all readers are not experts.

2.      In table 1, the use of medications (yes or not). It is very confusing to know for example, if 69.0 patients out of 142 (having no or mild NAFLD) use metformin or not and so on.

Author Response

The authors have revised the points mentioned in the comments however, the mixing of several parameters has made it difficult to believe in the conclusion.

Question 1: The participants had smoking and drinking habits and had a mixture of diseases that could be the reason for gut dysbiosis and alteration in SCFs regulation.

Answer 1: Thanks for your suggestion. This study is aimed to identify the relationship between SCFAs and NAFLD severity in T2D patients. The severity of NAFLD is the main outcome in this study. Smoking, alcohol drinking, and medical disease may be related to SCFAs regulation and NAFLD. To reduce the effect of possible confounding factors on the relationship between circulating SCFAs levels and the severity of NAFLD, the covariates including clinical data, laboratory data, and medication are initially analyzed in univariate analysis. Then multivariate logistic regression models including significant variables (p < 0.05) in univariate analysis and other traditional covariates, such as habit of smoking, alcohol drinking, medical disease, are adjusted to clarify the associations between plasma levels of SCFAs and severity of NAFLD. In univariate analysis, BMI ≥ 27 kg/m2, metformin usage and serum GOT, GPT, hemoglobin (Hb), log-formed TG, and isovalerate levels were significantly and positively correlated to an elevated risk of moderate to severe NAFLD in T2D patients. Smoking, alcohol drinking, hypertension, gout, and hyperlipidmia are not significantly related to NAFLD severity in T2D patients. After adjusting for age, BMI ≥ 27 kg/m2, T2D duration, Hb, GOT, GPT, log-formed TG, metformin usage, habit of smoking, alcohol drinking, history of hypertension, gout, and hyperlipidemia, the T2D patients with low levels of circulating isobutyrate and methylbutyrate levels had an increased risk of moderate to severe NAFLD. Please see Table 1, Table 2, 2.5 Statistical analysis in page 3, and 3.3. Circulating SCFA levels and the NAFLD severity in page 6

Question 2: The use of different drugs is another parameter which is not normalized in all participants.

Answer 2: Thanks for your suggestion. If the use of different drugs is normalized, it leads to selection bias. Our findings found no significant relationship between most drug use and NAFLD severity in T2D patients excluding merformin. After adjusting significant variables, metformin and traditional covariates, a negative association between circulating levels of isobutyrate and methylbutyrate and NAFLD severity in T2D subjects was still found. Please see Table 1, Table 2, 2.5 Statistical analysis in page 3, and 3.3. Circulating SCFA levels and the NAFLD severity in page 6, and Discussion, page 9, 5th paragraph

Question 3: The authors have mentioned the lack of controls in the limitations of the study that indicates the potential false conclusion.  

Answer 3: Thanks for your suggestion. There are no non-diabetic controls in this study to compare with T2D patients. However, our study aim is to examine the relationship between circulating SCFAs and NAFLD severity in T2D patients, not in general population. Thus, these findings were not influenced no matter whether non-diabetic individuals were enrolled in this study. Please see Discussion, page 10, second paragraph.

Moreover, have a look at two minor points mentioned in the manuscript

Question 4:   Write the full word of any abbreviation when use first time in the manuscript. Like SI mix solution (Does IS mean internal standard or International standard?). An expert can know however, all readers are not experts.

Answer 4:  Thanks for your correction. We have added full word of abbreviation. IS mix solution mean internal standard mix solution, and please see page 3 and 3rd paragraph.

Question 5: In table 1, the use of medications (yes or not). It is very confusing to know for example, if 69 patients out of 142 (having no or mild NAFLD) use metformin or not and so on.

Answer 5: Thanks for your correction. We have corrected it to percentage (%) and revised Table 1. 

Reviewer 2 Report

1. What is the main question addressed by the research? PARTIALLY
2. Do you consider the topic original or relevant in the field? Does it
address a specific gap in the field? NOT SIGNIFICANT
3. What does it add to the subject area compared with other published
material? A negative association between
circulating levels of isobutyrate and methylbutyrate were associated in type-2 diabetic subjects with NAFLD severity.
4. What specific improvements should the authors consider regarding the
methodology? What further controls should be considered? Look for novel targets and biomarkers
5. Are the conclusions consistent with the evidence and arguments presented
and do they address the main question posed? Partially
6. Are the references appropriate? Yes
7. Please include any additional comments on the tables and figures. Not applicable

Author Response

Question 1: What is the main question addressed by the research? PARTIALLY

Answer 1: Thanks for your comment. This study was aimed to explore the association between SCFAs with the severity of NAFLD in T2D patients.

Question 2: Do you consider the topic original or relevant in the field? Does it
address a specific gap in the field? NOT SIGNIFICANT

Answer 2: Thanks for your comment.

Question 3: What does it add to the subject area compared with other published
material? A negative association between circulating levels of isobutyrate and methylbutyrate were associated in type-2 diabetic subjects with NAFLD severity.

Answer 3: Thanks for your comment.

Question 4: What specific improvements should the authors consider regarding the
methodology? What further controls should be considered? Look for novel targets and biomarkers

Answer 4: Thanks for your comment. There are no non-diabetic controls in this study to compare with T2D patients. However, our study aim is to examine the relationship between circulating SCFAs and NAFLD severity in T2D patients, not in general population. Thus, these findings were not influenced no matter whether non-diabetic individuals were enrolled in this study. Please see Discussion, page 10, second paragraph.

Further study is necessary to explore novel targets and biomarkers of NAFLD in T2D population in the future. We added this description in Discussion (page 10 and 2nd paragraph)

Question 5: Are the conclusions consistent with the evidence and arguments presented
and do they address the main question posed? Partially

Answer 5: Thanks for your comment

Question 6: Are the references appropriate? Yes
Answer 6: Thanks for your comment

Question 7: Please include any additional comments on the tables and figures. Not applicable

Answer 7: Thanks for your comment

Round 3

Reviewer 2 Report

The manuscript is now significantly improved and may be accepted